# Another Way to Confuse Motor Control: Manual Technique Supposed to Shorten Muscle Spindles Reduces the Muscular Holding Stability in the Sense of Adaptive Force in Male Soccer Players

**DOI:** 10.3390/brainsci13071105

**Published:** 2023-07-21

**Authors:** Frank N. Bittmann, Silas Dech, Laura V. Schaefer

**Affiliations:** 1Regulative Physiology and Prevention, Department Sports and Health Sciences, University of Potsdam, 14476 Potsdam, Germany; 2Health Education in Sports, Department Sports and Health Sciences, University of Potsdam, 14476 Potsdam, Germany

**Keywords:** maximal isometric Adaptive Force, holding capacity, muscle stability, muscle instability, neuromuscular functioning, neuromuscular control, motor control, muscle spindle, muscle physiology, regulatory physiology

## Abstract

Sensorimotor control can be impaired by slacked muscle spindles. This was shown for reflex responses and, recently, also for muscular stability in the sense of Adaptive Force (AF). The slack in muscle spindles was generated by contracting the lengthened muscle followed by passive shortening. AF was suggested to specifically reflect sensorimotor control since it requires tension-length control in adaptation to an increasing load. This study investigated AF parameters in reaction to another, manually performed slack procedure in a preselected sample (n = 13). The AF of 11 elbow and 12 hip flexors was assessed by an objectified manual muscle test (MMT) using a handheld device. Maximal isometric AF was significantly reduced after manual spindle technique vs. regular MMT. Muscle lengthening started at 64.93 ± 12.46% of maximal voluntary isometric contraction (MVIC). During regular MMT, muscle length could be maintained stable until 92.53 ± 10.12% of MVIC. Hence, muscular stability measured by AF was impaired after spindle manipulation. Force oscillations arose at a significantly lower level for regular vs. spindle. This supports the assumption that they are a prerequisite for stable adaptation. Reduced muscular stability in reaction to slack procedures is considered physiological since sensory information is misled. It is proposed to use slack procedures to test the functionality of the neuromuscular system, which is relevant for clinical practice.

## 1. Introduction

After decades of research, the detailed origin of injuries to joints, muscles, or connective tissue due to physical activity in sports or everyday life seems to be insufficiently clarified—especially when there is no overwhelming external impact to the body by another person or object. It was suggested that non-contact injuries occur when muscles try to slow down movements caused by inertial forces, which are often enhanced by additional gravitational effects [1,2,3,4]. The muscular stabilization under load plays an important role in keeping the articulating surfaces of joints aligned with each other. To provide such protective stabilization, an adequate adaptation of the neuromuscular action to the actual load is necessary. Especially if the intensity of the load changes quickly, the stabilizing control mechanisms must meet the requirements precisely and immediately in time. That specific adaptation of the neuromuscular system to an external force has not been sufficiently considered in motor science so far. Due to its adaptive character, we have introduced the term Adaptive Force (AF) for it [5,6,7,8,9,10,11,12,13,14]. Research over the last few years has shown that this muscle function can be understood as a unique kind of strength. As described previously, the “AF not only requires muscle strength but also sensorimotor control. It reflects the neuromuscular functionality to adapt adequately to external forces with the intention of maintaining a desired position or movement” [6]. The maximal isometric AF (AFiso_max_) is the decisive parameter that characterizes the maximal adaptive holding capacity. It stands for the maximal force a muscle can generate under static conditions against an increasing load. If the muscle starts to lengthen during the force rise (surpassing AFiso_max_), the stable position is left, and the muscle merges into eccentric action. In this phase, the force usually increases further until the maximal eccentric AF (AFecc_max_) is reached.

The specific motor action of AF relies, amongst other things, on a well-functioning proprioceptive system, including muscle spindle afferents. Muscle spindles are most important for kinaesthesia where additional information is provided, e.g., by skin receptors [15]. The Information running from muscle spindles to the central nervous system relates to muscle length and limb position, changing the status of muscle tone and movement [16]. Macefield & Knellwolf provided an elaborate overview of the topic [17].

The functionality of muscle spindles can be impaired when their sensitive parts get slack under certain conditions. It is commonly accepted that a slack in muscle spindles can occur [18,19]. The group around Proske used a slack-procedure that included a short contraction of the muscle in lengthened position followed by a passive shortening of the muscle to middle test position (named CL-procedure in the following). It was suggested that the intrafusal fibers are unable to readjust after passive return and fall slack [20]. This procedure led to a significant reduction in the stretch reflex [21]. The effect was revoked if the muscle was contracted again at test length prior to the reflex test. The authors presumed that the second contraction tightened the slacked muscle spindles again [21]. Since research on shortened or slacked muscle spindles is mainly aimed at reflex reactions [15,17,21,22,23,24], we recently performed a study investigating the behavior of AF after such procedures [14]. The AF was assessed by an objectified manual muscle test (MMT). The maximal holding capacity was almost halved after the CL-procedure compared with regular testing (without any conditioning): the muscles started to lengthen already at ~53% of the maximal force, which was then reached during eccentric motion. This substantial and highly significant effect was cancelled by the short second contraction in the test position [14]. Furthermore, the peak value of the trial (AF_max_) was still reached under isometric conditions (AFiso_max_ ≈ AF_max_) as for regular MMTs. Hence, the difference was not the magnitude of maximal force but the type of motor action in which it was reached—during static conditions or muscle lengthening. The assumed slack led obviously to an immediate, clearly, and highly significant impairment of the adaptive holding capacity in a given steady muscle length in reaction to an increasing external load. The results suggest that muscle history cannot only influence experimental reflex testing [21] but also the muscular holding capacity, which is assumed to be important for the stabilization of musculoskeletal structures during motions in sports and daily life.

On that basis, the question arises if other procedures could alter the functionality of muscle spindles. A manual spindle technique is used in Applied Kinesiology (AK) [25,26]. For that, the muscle area in which muscle spindles are supposed to be located is pressed together following the longitudinal axis of muscle fibers [26]. It is hypothesized that this manipulates sensory information by passively shortening the muscle fibers which would lead to a slack in the muscle spindles. Presumably, other mechanosensors of the skin and fascia would be involved, too. This manual spindle technique is used in AK in combination with the MMT for diagnostic purposes [26].

The MMT performed as a “break test” [27] (see Section 2.3) can be used to assess the AF. It enables a flexible, time-saving assessment and is close to clinical practice. Two qualities can be differentiated thereby, which are assessed by the tester’s feelings during the test: (1) the MMT is rated as “stable”, if the participant is able to maintain the isometric position up to a considerably high force level despite the external force increase applied by the tester. Thus, the muscular interaction of tester and participant remains under static conditions during the whole MMT. (2) The MMT is rated as “unstable”, if the participant’s muscle starts to lengthen during the external force increase; thus, the static condition is left and the muscle merges into eccentric action. In clinical practice, a muscle is considered “normoreactive” [26] if a formerly stable MMT turns out to be unstable after the manual spindle technique. From a sensorimotor point of view, this reaction can be considered physiological since the sensory information of the muscle spindles will probably not match the overall muscle length in case the muscle spindles fall slack. However, this is based on clinical experiences; quantitative data are missing.

The aim of the present study was to investigate if this phenomenon could be quantified by the AF. AF parameters were assessed by an objectified MMT in reaction to the manual spindle technique compared with regular MMT (without preceding spindle manipulation). The MMT was objectified by a handheld device that records force and limb position simultaneously [8,9,10,11,12,13,14]. Based on the results of the previous study [14], it was hypothesized that the holding capacity (AFiso_max_) would be significantly reduced after spindle technique, whereas the AF_max_ would not be affected. Additionally, the onset of oscillations in the force signal during the MMT was expected to be on a significantly higher force level following spindle technique compared with regular MMT.

## 2. Materials and Methods

The measurements were performed at one appointment at the Neuromechanics Laboratory of the University of Potsdam or at the practice of Integrative Medicine Bittmann (both Potsdam, Germany). All MMTs of elbow and hip flexor muscles were executed by one experienced male tester (m, 65 years, 185 cm, 87 kg, 27 years of MMT experience) and were objectified by a wireless handheld device. Although the setting (test position and test vector; see below) was aligned to mainly activate the biceps brachii and rectus femoris muscles, synergists are supporting the muscle action. Therefore, the two tested muscle groups will be termed elbow flexors and hip flexors in the following.

### 2.1. Participants

A priori sample size estimation was performed with G*power (version 3.1.9.7, Düsseldorf, Germany). The intra-individual comparison of AF parameters, especially AFiso_max_, between regular MMT and MMT after manual spindle technique is of the utmost relevance. The previous study using the CL-procedure to generate slacked muscle spindles showed a very large effect size for AFiso_max_ of *d_z_* = 1.8 [14]. Since this study evaluated a manual spindle technique and another sample (soccer players), the estimation for a paired *t* test was performed using a smaller, but still large effect size of *d_z_* = 0.9 (two-tailed, α = 0.05, 1 − β = 0.8). This revealed a minimal sample size of n = 12.

Thirteen healthy males volunteered to participate in this study (age: 26.38 ± 4.25 years (range: 19–34), body mass: 77.77 ± 4.78 kg, body height: 179.85 ± 5.57 cm). To ensure a homogenous group regarding gender and activity level, only male semi-professional soccer players (regional league) were included. Exclusion criteria were any current complaint or injury of the measured extremities. The inclusion criterion was a stable neuromuscular function of the measured muscles (see Section 2.5). Both muscle groups (elbow and hip flexors) on one side were measured in ten participants. In the remaining three participants, only one limb (elbow or hip flexors) could be measured due to impaired neuromuscular function in the other one. In total, 11 elbow flexors (n = 10 right, n = 1 left) and 12 hip flexors (n = 8 right, 4 = left) were examined and considered for evaluation.

The study was conducted according to the guidelines of the Declaration of Helsinki and was approved by the Ethics Committee of the University of Potsdam, Germany (protocol code 35/2018; 17 October 2018). All participants gave their written, informed consent to participate.

### 2.2. Technical Equipment

The used wireless handheld device was described previously (Figure 1a) [8,9,10,11,12,13,14]. Integrated strain gauges (model: a14071900ux0076, precision: 1.0 ± 0.1%, sensitivity: 0.3 mV/V; co. Sourcing Map, Hong Kong, China) recorded the reaction force and kinematic sensors (Bosch BNO055, 9-axis absolute orientation sensor, sensitivity: ±1%; Bosch Ltd., Stuttgart, Germany) the angular velocity simultaneously during the MMTs with a sampling rate of 180 Hz. Data were AD converted and transmitted (Bluetooth) to a tablet (Sticky notes, StatConsult Ltd., Magdeburg, Germany).

### 2.3. Setting and Manual Muscle Tests

For all tests, the participant was placed in a supine position. Figure 1b displays the group test for elbow flexors, and Figure 1c for hip flexors. Due to the adjustment of the respective limb and the used test vector, the biceps brachii muscle should be mainly activated for the elbow flexor test (90° flexion, maximal supination of the forearm) and the rectus femoris muscle for the hip flexor test (90° hip and knee flexion). Both test vectors are aligned with the direction of the respective muscle fibers. The tester had contact with the distal part of the forearm or thigh, respectively. The contact point was marked to ensure an equal intra-individual point of force application. To measure the dynamics and kinematics during the tests, the handheld device was located between the tester’s palm and the participant’s limb (Figure 1b,c).

For AF measurements, MMTs were performed in the sense of a break test using the handheld device for objectification, as previously described [9,10,11,12,14]. The participant’s task was to maintain the test position (Figure 1b,c) while the tester applied an increasing force to the participant’s limb (S-shaped force rise as depicted by Bittmann et al., [8] (p. 25)). Hence, the participant had to adapt to the applied force of the tester by holding isometrically (holding isometric muscle action; HIMA). If the participant was able to maintain the test position up to a considerably high force level, the adaptation is considered adequate (tester’s rating: stable). In the event that the limb gave way during the force rise, the neuromuscular function is interpreted as impaired (tester’s rating: unstable). Thereby, the initial HIMA merged into eccentric muscle action in the course of the test. The force is usually increased further during that muscle-lengthening phase.

For MVIC tests, the same test position was adjusted, but the participant should now push as strong as possible against the stable passive resistance provided by the tester [14]. Hence, the participant performed a pushing isometric muscle action (PIMA) [28].

### 2.4. Manual Spindle Technique: Manipulation of Muscle Spindles

The experimental condition was a manipulation aimed at shortening the regarding muscle spindles (termed “manual spindle technique”). For the reason that the manipulation is applied via the skin, it is likely that skin receptors are also involved. The tester positioned the thumbs and index fingers of both hands pointing in opposite directions above the muscle belly in a longitudinal axis with the muscle fibers (biceps brachii or rectus femoris muscle, Figure 1d). In that position, he pressed his fingers into the skin and then pushed them together so that the skin and underlying tissue, including the muscle belly, were shortened in the direction of muscle fibers (Figure 1e). The procedure was performed with quick and intense impulses and was repeated six times within ~2 s. Directly afterwards, the MMT of the respected muscle was executed. It is assumed that by using this manual technique, the muscle spindles are shortened passively, and an intrafusal slack would result in the short term.

### 2.5. Procedure

After the participants were introduced to the measurement procedure and signed the informed consent form, the suitability of the elbow and hip flexors of the preferred/dominant side of each participant was checked clinically (without a handheld device). One exclusion criterion was specified above as impaired neuromuscular functioning. An appropriate neuromuscular function is clinically determined by two characteristics: (1) the MMT must be initially stable, and (2) it should get unstable after the manual spindle technique [26]. It is known from practical experience that muscles do not always show this behavior; thus, this had to be tested in advance of the AF measurements. Since the aim of the study was not to investigate if the manual spindle technique leads to unstable MMTs but how the AF parameters behave, this procedure seems appropriate. Limbs were excluded from measurements if the neuromuscular function of the tested elbow or hip flexors was not appropriate (unstable or no reaction to spindle technique) or if any pain occurred during the MMT. In those cases, the other side was checked for suitability. In case both sides had to be excluded, the regarding limb was not considered in this subject. Given that the other limb (hip or elbow) fulfilled the requirements, it was measured. This led to the number of included limbs (11 elbow flexors, 12 hip flexors, n = 13 participants).

Subsequently, in total, 8 measurements were performed per limb. The measurement series started with two MVIC tests of each limb for reference (alternating order of limbs, starting with hip flexors). Subsequently, the AF measurements were performed starting with hip flexors: 1. MMT after manual spindle technique and 2. regular MMT for reference (without manipulation). Two trials of elbow flexors followed (1. spindle, 2. regular). This approach was repeated twice, so that three measurements of each limb (elbow and hip) and procedure (spindle vs. regular) were recorded. The resting period between the trials was 60 s.

### 2.6. Data Processing and Statistical Analyses

Force and gyrometer signals were analyzed as described previously using NI DIAdem 2017 (National Instruments, Austin, TX, USA; for details see [9,10,11,12,14]). Equidistant time channels were achieved by interpolating signals (linear spline) at a sampling rate of 1000 Hz. Subsequently, a Butterworth low-pass filter was applied (cut-off frequency 20 Hz, filter degree 5). The following parameters were extracted (all in N):MVIC: peak value of the two MVIC tests.AF_max_: peak value of each MMT trial irrespective of the occurred muscle action (could either be reached under static conditions (stable MMT) or during muscle lengthening (unstable MMTs)). For the former, AF_max_ = AFiso_max_. For the latter, AF_max_ = AFecc_max_ > AFiso_max_ (Figure 2).AFiso_max_: highest force value under static conditions, thus, during the isometric holding phase of an MMT. Since this does not necessarily refer to a peak value, the gyrometer signal was used to detect a potential breaking point (limb movement in the direction of joint extension). If the angular velocity increased consistently above zero (indicating yielding), the force value at the last zero crossing was referred to as AFiso_max_. If static conditions were present until the peak value was reached, AFiso_max_ = AF_max_. (Figure 2; for a detailed description, see [9,10,11,12,14]).AFosc: force at the onset of oscillations, referring to a clear upswing of subsequent oscillations in the force signal (for details see [9,10,11,12,14]). If no clear upswing arose, AFosc = AF_max_.

The ratios $\frac{{AFiso}_{max}}{{AF}_{max}}$, $\frac{AFosc}{{AF}_{max}},\frac{{AFiso}_{max}}{{AF}_{osc}}$ as well as $\frac{{AFiso}_{max}}{MVIC}$ were additionally computed. Furthermore, the slope of force rise should be considered when comparing the regular MMT vs. the MMT after spindle technique. For that, the signals were smoothed (width: 250 points). The slope was calculated by the difference quotient using time and force values of 40% and 70% of the minimal AFiso_max_ value of all MMTs of one limb per participant, which were rated as unstable. Due to the exponential behavior of slope, the logarithm was taken [lg(N/s)].

The arithmetic means (M), standard deviations (SD), and 95% confidence intervals (CI) were calculated for all AF parameters. Additionally, the intraindividual coefficient of variation (CV_intra_) was computed. It stands for the arithmetic mean of the participants’ CVs for the trials that were assessed as stable or unstable.

Statistical comparisons were completed using SPSS Statistics 29 (Windows, Version 28.0., IBM Corp., Armonk, NY, USA). All parameters were normally distributed (Shapiro-Wilk-test) except for $\frac{{AFiso}_{max}}{{AF}_{max}}$ of regular MMTs for both limbs, $\frac{{AFiso}_{max}}{{AF}_{osc}}$ of regular MMTs (hip flexors) and after spindle technique (elbow flexors), as well as the slope of elbow flexors. In cases where normal distribution was confirmed, pairwise comparisons between regular vs. spindle techniques were performed by paired *t* tests. For non-parametric data, the Wilcoxon-U-test was executed. Effect sizes Cohen’s *d_z_* for *t* test and Pearson’s *r* = $\left|\frac{z}{\sqrt{n}}\right|$ for Wilcoxon test were given and interpreted as “small” (0.2), “moderate” (0.5), and “large” (0.80) [29]. For *d_z_* a value of 1.3 stands for a “very large” effect [29]. The significance level was set at α = 0.05.

## 3. Results

Table 1 displays the values of the MVIC and AF parameters of the elbow and hip flexors and the statistical values comparing regular vs. spindle procedures. As a preliminary consideration, the slope of force increase should be considered. The exemplary curves (Figure 2) depict similar force rises. The slope did not differ significantly between regular vs. spindle procedures for both limbs (Table 1, Figure 3).

### 3.1. Parameters of Adaptive Force Comparing Regular vs. Spindle Procedure

The most relevant parameter comparing regular vs. spindle was the highest AF under isometric muscle action (static conditions). AFiso_max_ was significantly lower for spindle vs. regular for both limbs with very large effect sizes (*d_z_* > 2.8) (Table 1, Figure 4a,d). In contrast, the peak force (AF_max_) was significantly higher after spindle vs. regular for both muscles (Table 1, Figure 4b,e). It must be pointed out that AF_max_ was gathered mainly under static conditions for regular MMTs but under yielding conditions (muscle lengthening) in all 69 trials after manual spindle technique. This is reflected by the ratio AFiso_max_/AF_max_ (Table 1, Figure 4c,f and Figure 5).

Figure 5 illustrates the ratio AFiso_max_/AF_max_ for all single trials. Comparing regular vs. spindle, 81.82% vs. 0% of all trials of elbow flexors (n = 33) and 94.44% vs. 0% of all trials of hip flexors (n = 36) showed a ratio higher than 95%. A ratio between 80–95% was achieved for elbow flexors in 15.15% (regular) vs. 6.06% (spindle) of all trials and in 0% vs. 11.11% for hip flexors. Accordingly, in 3.03% vs. 93.04% (elbow) and 5.55% vs. 88.89% (hip) of all trials of regular vs. spindle, the ratio was below 80%. For manual spindle technique, 39.13% of all 69 trials (elbow and hip together) revealed a ratio even below 60%. Only one participant (no. 11, Figure 5) showed a ratio lower than 60% in the last trial with regular procedure of hip flexors. For this, the MMTs were rated as “unclear” by the tester, and the participant reported feeling exhausted.

Summarizing these results, the participants started to yield at a significantly lower force level after manual spindle technique (AFiso_max_/AF_max_ = 62.6 ± 10.2%, average for both limbs), where the peak force was reached during muscle lengthening for all trials. For regular MMTs, the peak value was reached under static conditions for the vast majority of trials (AFiso_max_/AF_max_ = 97.5 ± 5.9%).

In general, the force increased further in case of muscle lengthening. AF_max_ was significantly higher for spindle vs. regular procedure (Table 1, Figure 4a,d). This might have skewed the values of the ratio AFiso_max_/AF_max_ in the direction of the trials after the manual spindle technique. Hence, the maximal holding capacity should also be related to the MVIC. The participants reached 91.9 ± 10.3% (elbow) and 93.2 ± 10.4% (hip) of the MVIC for regular MMTs, which differed not significantly between AFiso_max_ and MVIC (elbow: t (10) = 0.961, *p* = 0.359; hip: t (11) = −0.100, *p* = 0.924). Following manual spindle technique, the relation AFiso_max_/MVIC decreased to 62.6 ± 12.3% (elbow) and 67.0 ± 12.8% (hip) of MVIC (Table 1). Consequently, the ratio AFiso_max_ to MVIC differed significantly between manual spindle technique and regular MMT with very large effect sizes of *d_z_* > 2.2 (Table 1).

### 3.2. Onset of Oscillations Comparing Regular vs. Spindle Procedure

The onset of oscillations occurred at a clearly lower force level for regular MMT vs. MMT after manual spindle technique for both limbs (Table 1), with very large effect sizes of *d_z_* > 2.6. The upswing started at a 20.4 ± 8.3% (elbow) and 20.7 ± 10.5% (hip) higher level for spindle vs. regular. After manual spindle technique, the oscillations arose regularly after the breaking point (ratio AFosc/AFiso_max_) for both limbs (elbow: 33 of 33 trials, hip: 34 of 36 trials) (Table 1). For regular MMT, the oscillations occurred before AFiso_max_ was reached, at an average of 86.4% (elbow) and 80.5% (hip) of AFiso_max_ (Table 1, elbow: 30 of 33 trials, hip: 34 of 36 trials). The ratios AFosc/AF_max_ and AFosc/AFiso_max_ were also significantly higher for spindle vs. regular (Table 1).

### 3.3. AF Data in Relation to the Tester’s Ratings of Manual Muscle Tests

The tester rated all 33 regular MMTs of elbow flexors and 35 of 36 regular MMTs of hip flexors as stable (one hip flexor test was assessed as unclear). All MMTs after the spindle technique were assessed as unstable (Table 2).

24 (elbow) and 33 (hip) of the MMTs that were assessed as stable showed ratios of AFiso_max_/AF_max_ = 100%, thus being under completely static conditions (Figure 6a). This indicates the participant was able to perfectly adapt their muscle tension in isometric position to the tester’s force increase until the maximal force was reached. As Figure 6b depicts, seven MMTs of elbow flexors (21%), which were rated as stable, showed values < 98% (M ± SD: 86.41 ± 6.42%; range: 78.17–96.12%); for hip flexors, this accounted for two MMTs (96.66% and 70.46%). For all MMTs that did not show ratios of 100% even though they were rated as stable, the tester reported higher suspensions during the test than usual. A closer inspection showed that the gyrometer signal did not clearly increase above the zero line; it stayed close to zero and came back to zero after the maximal value was reached (Figure 7). This presumably reflects the higher suspension that the tester perceived. Due to the algorithm, the AFiso_max_ value had to be determined, nevertheless, at the first zero crossing of the gyrometer signal, after which it did not return below zero until the maximal value was reached. This might highlight special cases that have to be distinguished from the clearly yielding MMTs. This will be discussed in the limitations section. 

For MMTs that were rated as unstable, 94% (elbow) and 89% (hip) showed ratios of AFiso_max_/AF_max_ < 80% (Figure 6). It is noteworthy that in 22% (elbow) and 21% (hip) of unstable MMTs, the AFiso_max_ was halved relative to AF_max_ (Figure 6); in four elbow flexor tests, the holding capacity was reduced even below 40% of AF_max_. Six MMTs (2 elbow, 4 hip) revealed ratios higher than 80%. Especially for one hip flexor test, the ratio was considerably high at 94.60%, although the tester rated the MMT as unstable. Here again, the curves showed a special behavior since the gyrometer signal left the zero line but returned for two oscillations below zero before the maximal force was reached. Hence, the AFiso_max_ value had to be determined at this high force level.

Furthermore, it has to be pointed out that the CV_intra_ of AFiso_max_/AF_max_ was considerably higher for unstable vs. stable MMTs (Table 2). This was especially due to the high CV_intra_ of AFiso_max_ for unstable vs. stable MMTs (averagely 23% vs. 7% for elbow and 15% vs. 6% for hip flexors, respectively), where the CV_intra_ of the AF_max_ values was averagely 6% vs. 4% for elbow and 7% vs. 4% for hip flexor tests, respectively.

Overall, the tester’s ratings are supported by the quantitative results.

## 4. Discussion

The objective of this study was to investigate the effect of a manual spindle technique on parameters of AF assessed by an objectified MMT compared with regular testing without manipulation in participants with “normoreactive” muscle function. The central result was that the manual spindle technique led to a significant reduction of the maximal holding capacity and, thus, an impairment of muscular stability. This was reflected by the clearly reduced AFiso_max_. In contrast, the maximal force reached during the test (AF_max_) was highest after spindle technique.

### 4.1. Methodological Considerations Comparing Regular vs. Spindle Procedure

The investigation included the manual assessment of the maximal holding capacity by a MMT, which was objectified by a handheld device. The force profile that is applied by the tester to the extremity of the participant during the MMT must meet some requirements. Firstly, the maximal force applied by the tester is relevant. The force maximum reached during one trial depends on the interaction of both partners (tester and participant), as previously described [14]. Under stable conditions (which were mostly present for regular MMTs), the maximal force during the test depends mainly on the tester’s applied maximal force. It was suggested previously that in cases where the tester’s force application would be too low, an actually unstable MMT could be falsely rated as stable [14]. The AFiso_max_ for regular MMTs showed by −8% and −7% lower values compared with the MVIC (not significant). Since the MVIC vs. AFiso_max_ of regular MMTs did not differ significantly, the applied maximal force of the tester during the MMTs can be considered appropriately high. In addition to the tester’s action, the maximal force under unstable conditions depends especially on the participant’s adaptive capacity. After manual spindle technique, the AFiso_max_ was significantly lower than the MVIC by −37% and −33% for the elbow and hip flexors, respectively. This indicates that after such a manipulation, the participants were not able to generate their maximal force under static conditions. The static condition was left, and during eccentric motion, the force increased further (occurred in all trials after manual spindle technique). The force rose to even significantly higher values compared with regular MMT (+10%) and MVIC (+5%). Although it is known from the previous studies [9,10,11,14] that the force increases further during muscle lengthening, it was not expected that the maximal force values would be significantly higher than AFiso_max_ ≈ AF_max_ of regular MMT, as found here in contrast to the previous studies. Some investigations showed higher forces during eccentric motions compared with isometric ones [30,31,32,33], which was suggested to be related, inter alia, to training status [34]. Hence, the reason for the higher AF_max_ for spindle vs. regular procedure could possibly be the examined sample of soccer players. Nevertheless, the decisive factor is that, after manual spindle technique, the maximal holding capacity was significantly reduced. Summarizing the aspect of maximal applied force, it can be concluded that all MMTs ran with at least approximately maximal intensities and, therefore, were appropriately high.

With respect to the applied force profile, secondly, the slope of force development must be considered. It was suggested previously that a fast increase could make it “more difficult for the tested muscle to lock into a stable resistance” [14] (p. 12). Since the slope of force rise in the linear section did not differ significantly between regular and spindle procedures, the clearly and significantly differing AFiso_max_ values should not have been a result of the manual force application.

### 4.2. Comparison of Tester’s Ratings of Manual Muscle Test and AF Data

The ratings of MMTs were supported by the AF data in the vast majority of trials. Considering elbow and hip flexors together, 59 of 68 stable MMTs showed an AFiso_max_ ≥ 98% of AF_max_. In 63 of 69 the MMTs that were assessed as unstable, a ratio AFiso_max_/AF_max_ below 80% was found. Hence, considering all 137 trials together, in 89.05% of trials, the AF data support the tester’s ratings unambiguously. In 6.57% (9 of 137 trials), the ratio showed rather low deviations from those preliminary set thresholds (stable between 90–98% and unstable between 80–90%). Six trials (4.38%) showed stronger differences (stable: <90%, unstable: >90%).

As mentioned in the results section, for all MMTs that were assessed as stable and showed values below 98%, the tester perceived stronger suspensions. Those were visible in the gyrometer signals, showing a stronger increase above zero but not a clear break-off as for unstable MMTs. Furthermore, for some unstable MMTs that revealed high values of AFiso_max_/AF_max_, the gyrometer signals showed deviating behavior, e.g., stronger but rather slow oscillations that fell below zero in the short term after the signal had already clearly left the zero line; hence, the AFiso_max_ value had to be determined at higher levels due to the late zero-line crossing. Obviously, those cases show special behavior, which might be the nature of some patterns during such a personal interaction. The sample of male soccer players might have also been a factor since they showed high force levels in general, a different training status, and a specific sports type (soccer) characterized especially by powerful actions.

Previous studies in healthy participants without a clear assignment to particular sports showed a similar but sharper picture [9,10,11]. In Schaefer et al. [9], the data of the three different studies (with different objectives but all using the objectified MMT) were taken together to characterize stable and unstable behavior. It was suggested that stable adaptation is present for AFiso_max_ ≥ 99% of AF_max_. This was supported by the investigation using the CL-procedure [14], where regular MMTs (all assessed as stable) showed AFiso_max_ = 99.7 ± 1%. A further differentiation of stable and unstable MMTs was not given there. The other three studies revealed considerably low values of ~56.11% for unstable MMTs but a high CV of 38.94% [9]. We assume that this high CV reflects neuromuscular specialties of the holding function in case of instability. Nevertheless, the algorithm might also influence this parameter, but this is considered to be rather low.

Another aspect of defining stable vs. unstable muscle function was the onset of oscillation in previous studies [9,10,11,14]. It was suggested that the onset of oscillations could be a prerequisite for stable adaptation since it occurred before AFiso_max_ was reached. For the present study, in 64 of 68 stable MMTs (considering elbow and hip together), the onset of oscillations occurred at a force level lower than AFiso_max_ with an averagely −19.15% (range: −38.06% to −5.15%). In the remaining four stable MMTs, AFosc occurred at slightly higher values than AFiso_max_ (+10.17%, range: 0.55% to 15.40%). For 67 of 69 unstable MMTs, AFosc arose after the breaking point, concrete, on a 55.91% higher level than AFiso_max_ (range: 0.02% to 386.34%). For the remaining two unstable MMTs (both hip flexors), the onset of oscillations occurred prior to the breaking point at a force level of −12.65% and −16.93%.

A future evaluation should include all currently available data on stable and unstable MMT in healthy participants to sharpen thresholds for assigning the trials to stable or unstable based on the AF data. Thereby, it should also be examined if the algorithm could be optimized to also capture special cases. Nevertheless, it might be possible that the AF data of some special cases are not related to the algorithm but simply show deviating behavior of the adaptive motor function.

The objective AF data supported the subjective tester’s ratings in the vast majority of trials. Hence, the subjective feeling of muscular stability during the MMT can be mirrored by quantitative data. The benefit is that the diagnostics based on the MMT in clinical practice might be supported by objective AF data, which could also verify the quality of the tester’s skills. Thereby, the MMT could be moved out of the corner of subjectivity.

### 4.3. Confusing or Impairing Sensorimotor Control: Neurophysiological Considerations and Practical Implications with Respect to the Adaptive Holding Capacity

The manual spindle technique is intended to passively shorten muscle spindles. Assuming this results in a slack of muscle spindles, an influence on sensorimotor control was suggested. Previous investigations showed an influence of suspected slacked muscle spindles on stretch reflex [15,21,22,23] and on muscular holding capacity [14] by another slack-procedure. Thereby, the slack was generated by contracting the muscle actively in a lengthened position, followed by a passive shortening to the middle test position (CL-procedure). Comparing the AF data of the present study with our previous investigation [14], it can be stated that the maximal holding capacity behaved similarly for both slack-procedures. The participants were not able to achieve their maximal force under stable conditions after both spindle procedures. Following manual spindle technique, the AFiso_max_ was by −39% (elbow) and −36% (hip) lower than the AF_max_ which was then reached during muscle lengthening. The previous study using CL-procedure showed an even higher reduction of averagely −47% [14]. Comparing the AFiso_max_ values between slack-procedure and regular MMT, the present study revealed a reduction of −31% (elbow) and –28% (hip) after manual spindle technique, where the previous study showed a reduction of –44% for elbow flexors after CL-procedure. This indicates, firstly, that both slack-procedures led to a clear and significant reduction of the holding capacity; however, the CL-procedure resulted in a stronger decrease than the manual spindle technique used here. This distinction could be attributed not only to the different procedures but also to the different samples. As mentioned above, soccer players might show specific conditions due to training status or the sport type.

Then again, the CL-procedure could have a stronger effect on motor control than the manually performed pushing together of skin and muscle belly. From a neurophysiological point of view, during the former, the sensorimotor system is involved in a more active and complex way: the adjustment to the lengthened position is completed by an active contraction, thus physiologically. Even though the subsequent shortening was completed passively, there must have been proprioceptive influences not only from all synergistic elbow flexors but also from other involved structures like antagonists, tendons, and the joint capsule. This might have resulted in a higher impact compared with the single and narrowly limited manipulation of one muscle belly. Nevertheless, in both cases, the afferent signals that are led from the manipulated muscle spindle to spinal and supraspinal areas indicate an inappropriate shortness of the related muscle.

As was discussed previously, “at least the thalamus, cerebellum, inferior olivary nucleus (ION), red nucleus, basal ganglia, cingulate cortex, and the sensorimotor cortices are involved in the complex processing of adaptive motor control” [11] (p. 11). This information is based on a number of articles [35,36,36,37,38,39,40,41,42,43,44,45,46,47,48,49,50,51,52,53,54,55,56,57,58,59,60,61,62,63,64,65,66,67,68,69,70,71,72,73,74,75]. In this complex network, the sensory information of muscle spindles is cross-charged. In the case of slacked muscle spindles, presumably a mismatch with the overall muscle length occurs, and readjustment would be necessary. From a physiological perspective, the muscle tension must be lowered in order to lengthen the muscle again to resolve the mismatch. In the event that an increasing load impacts during this adjustment, the adaptation cannot be appropriate anymore, and the muscle starts to yield. Since a muscle function as AF includes the adjustment of muscle tension and length in reaction to an external force, it seems particularly suitable to detect mismatches of such a target/actual comparison of sensorimotor control.

As mentioned above, it is known that a short contraction in the test position reverses the slack effect [14,21,24]. It was questioned why the MMT did not have the same effect in the sense that the muscle spindles sent the correct information about muscle length at the start of the MMT [14]. It was assumed that the participant had to execute “a follow-up control in response to a varying reference input”, where “the initial state of proprioceptive inputs influences the iterative processes of the adaptive control loop during the whole process of a running test” [14] (p. 13). A more detailed discussion is given in Bittmann et al. [14] and is excluded here due to redundancies.

Obviously, both slack-procedures can reduce the holding capacity. From a neurophysiological perspective, this can be considered physiological and reflects normal regulation. That is why practitioners of Applied Kinesiology use the manual spindle technique to test if a muscle is “normoreactive” [26]. From clinical experience, it is known that, under certain circumstances, muscles do not show this physiological behavior. Muscles could fail to switch to instability after such a manipulation. Reasons for this can only be assumed, which was not the aim of this study but could be investigated in the future.

It is suggested that the executed HIMA—especially at the start of the MMT—is of special importance to test for stimuli that could possibly alter the adaptive motor function. Only this includes the adaptation of muscle length and tension. During pushing actions—e.g., during MVIC assessment—the length control, including the adaptive component, is not challenged to that extent.

Meanwhile, several stimuli that could cause an impairment of the muscular holding function were identified scientifically in healthy participants: muscle spindle slack-procedures [14], negative imagery [9,10], and unpleasant odors [11]. Long COVID patients initially showed a significantly reduced holding capacity, which stabilized immediately after helpful treatment/recovery. On that basis, and from clinical experiences, it is hypothesized that the holding capacity in the sense of AF is reactive to different stimuli entering the complex neuromuscular control circuitries, which are specifically demanded during AF [8]. Provided the holding capacity reflects neuromuscular functioning, as was supposed previously, reduced holding capacity in response to passively shortened (slacked) muscle spindles can be interpreted as physiological. Taking this neurophysiological consideration into account, muscle spindle slack-procedures are suggested as one way to check if the neuromuscular system is reactive and well-functioning.

When muscles are unstable, as was shown during unpleasant imaginations or smells as well as in the long COVID state, irritation of the control circuitries seems to be present, which leads to a misalignment of the adaptive neuromuscular control. Providing neurological illness is ruled out, a reduced holding function of muscles should not be understood as pathological but rather as a physiological response to disturbing influences.

Adaptive actions are highly relevant for practical implications since, in daily activities and sports, the neuromuscular system needs to react and adapt to the surrounding circumstances. In cases of muscular instability in the sense of AF, joints and passive structures might not be guided appropriately anymore. Orthopedic conditions without structural damage are often classified as “overload-injuries”. In contrast, the complaints frequently appear without high loads. Based on the knowledge of functional muscular instability, the focus should switch from load to loadability. Reduced muscular stability must lead to a worsened ratio between them, thus increasing the risk of a relative overload even under moderate stress. It seems that not the load is the problem, but rather the reduced loadability. Studies should focus less on maximal strength but rather on muscular stability in the sense of AF.

### 4.4. Limitations

As was discussed previously, the MMT that was used to perform the measurements is subjective by nature [14]. The handheld device offers the opportunity to objectify the MMT. Nevertheless, the reproducibility of the tester’s force profile must be given with respect to scientific quality criteria to collect reliable data. The tester proved his ability to reproduce the force profile against stable resistance in a previous study [8].

In contrast to previous studies, some results of AFiso_max_ suggest restrictions in part regarding the used algorithms. Those appeared in 9 of all 68 MMT (13%) that were assessed stable, especially for elbow flexors. Although this reflects a minority of cases, this point should be considered. As described in Section 3.3, in those MMTs, the tester perceived higher suspensions during the test, and the gyrometer signal showed a specific pattern. This phenomenon was occasionally present in previous studies, but not in the amount found here. It is assumed that it is the result of the considerably high forces in this study, which led to stronger oscillations and suspensions during personal interaction. Since the cutoff frequency of filtering for the gyrometer signal was 20 Hz, the muscular oscillations would not be cancelled. This was appropriate for most MMT evaluations until now. Regarding the different sample of soccer players, it could be considered to sharpen the evaluation of AFiso_max_ for future investigations, especially with participants in power sports. The cases in which the gyrometer signal left the zero-line short term with a small amount must be interpreted differently than those in which it clearly increased above zero. It could be considered to revise the evaluation accordingly for future investigations, especially regarding participants performing power sports. Furthermore, in most cases, the evaluation of AFiso_max_ is considered appropriate, and the outliers refer to the minority. Despite those probable restrictions, the data showed a clear picture, which would presumably be even stronger with an adjusted evaluation.

It has to be critically reflected that the muscles were pre-tested regarding the muscle spindle technique. Only muscles that showed a reaction to this manipulation (switched from stable to unstable MMT) were included. Since the aim of the study was not to investigate if the technique works in general but how the AF parameters behave if it does, this procedure was considered appropriate. Furthermore, the assessment was not blinded. The same tester applied the spindle technique (or no intervention), executed the MMT, and made the rating. The tester was aware of the study’s questions. Therefore, a bias during the test execution or rating cannot be ruled out. However, a conscious or unconscious manipulation of the applied force profile would have been recorded by the handheld device.

Further studies should verify the results in an extended sample with regard to a different population, potentially other testers, and a setting that might include single-blinding, which will not be trivial to realize.

## 5. Conclusions

The present study showed that a manual spindle technique can also lead to a reduction of the muscular holding function. This extends the knowledge of sensorimotor control: not only monosynaptic reflexes can be altered by slacked muscle spindles—as was shown by other researchers—but also the muscular stability.

Since the reaction to spindle procedures is considered physiological, we suggest including the holding function to investigate if neuromuscular functioning works appropriately. Further research in healthy people and patients should uncover the underlying mechanisms. It is suggested to use the holding capacity as a supportive diagnostic approach to investigate one aspect of neuromuscular functionality and draw conclusions on muscular stability. Since the maximal force was not influenced, it is suggested to focus less on maximal strength but rather on muscular stability in the sense of AF in future studies investigating injury mechanisms or musculoskeletal complaints.

On that basis, it can be examined further to determine which stimuli could have an impairing but also supporting effect on muscular stability. Practical implications regarding health states and complaints, as well as injury mechanisms, could be gained. This is assumed to be relevant for musculoskeletal complaints but also for more systemically active factors such as mental stress and post-infectious diseases.

## Figures and Tables

**Figure 1 brainsci-13-01105-f001:**
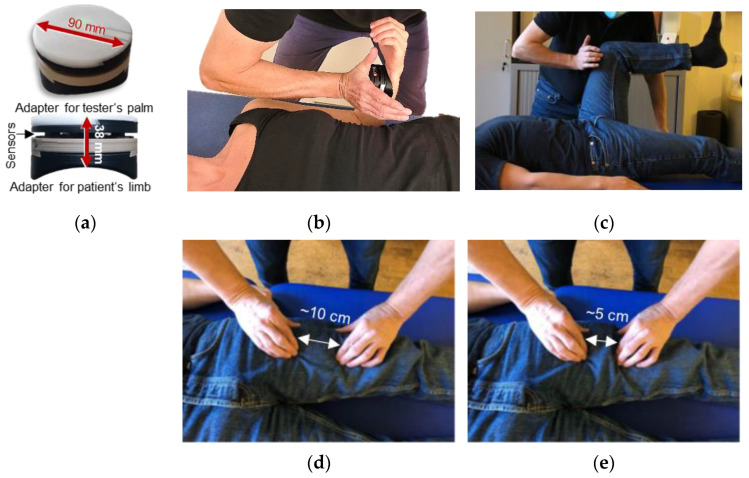
Equipment, setting and spindle manipulation. (**a**) Handheld device (reprint of [8,9,10,11,12,14]. Starting (test) position of the MMT of (**b**) elbow flexors and (**c**) hip flexors. (**d**,**e**) depict the start and end position, respectively, of the manual spindle technique exemplarily for the rectus femoris muscle as one of the main hip flexors regarding the performed test. This was analogously performed for the belly of biceps brachii muscle (elbow flexor test).

**Figure 2 brainsci-13-01105-f002:**
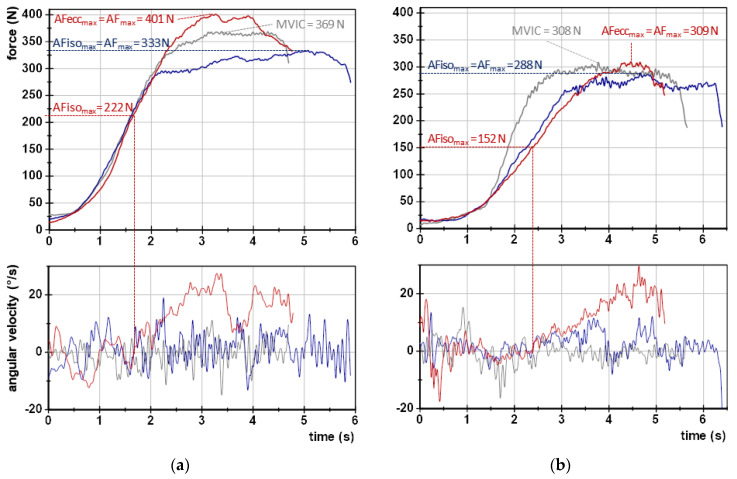
Exemplary curves of force and angular velocity. Displayed are both signals regarding one MVIC test (grey), one regular MMT (blue) and one MMT after spindle technique (red) for (**a**) elbow flexors and (**b**) hip flexors of the same participant (male, 24 years, 77 kg, 1.83 m). The values of MVIC, max. Adaptive Force (AF_max_) and max. isometric AF (AFiso_max_) (all in N) are given. Signals were filtered (Butterworth, low-pass, cut-off: 20 Hz, filter degree 5).

**Figure 3 brainsci-13-01105-f003:**
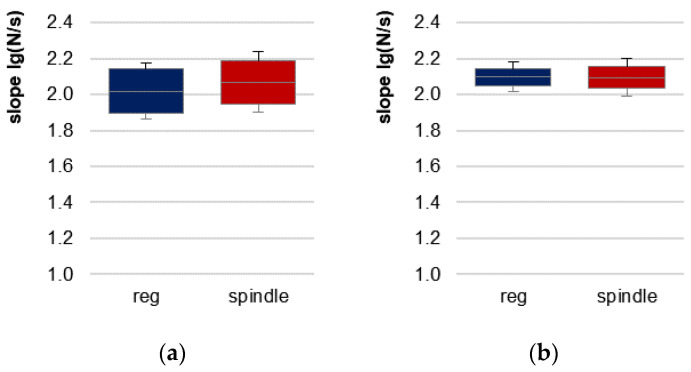
95%-confidence intervals including arithmetic means and standard deviations (error bars) of the slope of force increase [lg(N/s)] for regular MMT (reg, blue) and MMT after manual spindle technique (spindle, red) for (**a**) elbow flexors and (**b**) hip flexors. Statistical comparisons for both muscles were non-significant.

**Figure 4 brainsci-13-01105-f004:**
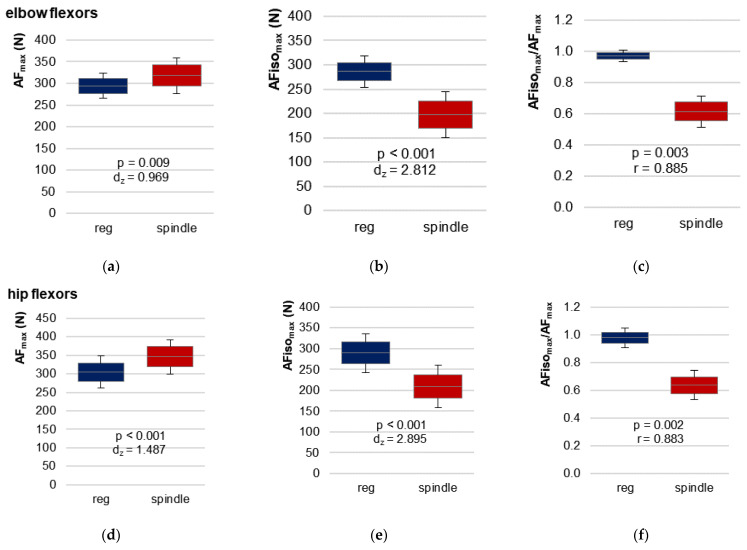
95%-confidence intervals including arithmetic means and standard deviations (error bars) of the maximal AF (AF_max_) (**a**,**d**), the maximal isometric AF (AFiso_max_) (**b**,**e**), and their ratio (**c**,**f**) for regular MMT (reg, blue) and MMT after manual spindle technique (spindle, red) for elbow flexors (**a**–**c**) and hip flexors (**d**–**f**). The *p*-values of pairwise comparisons and effect sizes Cohen’s *d_z_* (paired *t* test) or Pearson’s r (Wilcoxon test) are given.

**Figure 5 brainsci-13-01105-f005:**
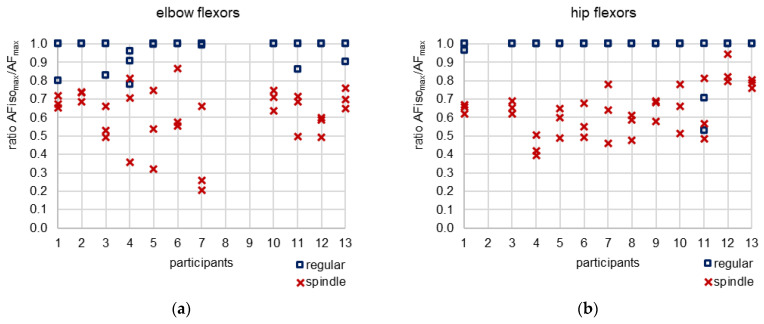
Single values of the ratio AFiso_max_ to AF_max_ for regular MMTs (blue square, three trials) and for MMTs after manual spindle technique (red cross, three trials) for each participant. (**a**) elbow flexors, (**b**) hip flexors. In case three trials are not visible the values were identical (applied especially for regular MMTs).

**Figure 6 brainsci-13-01105-f006:**
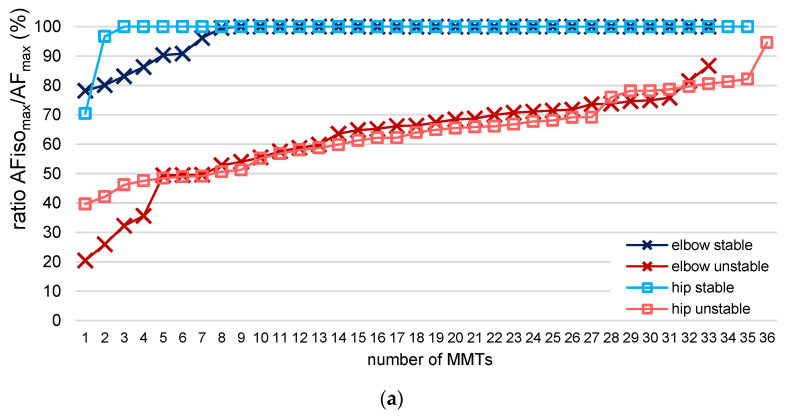
(**a**) Ratios AFiso_max_ to AF_max_ of all MMTs of elbow (cross) or hip flexors (square) which were rated as stable (dark/light blue) or unstable (dark/light red) sorted in ascending order. (**b**) Percentage of the total number of trials for which AFiso_max_/AF_max_ was between 98–100%, 90–98%, 80–90%, 70–80%, 60–70%, or <60% for MMTs which were rated as stable (blue) and unstable (red) for elbow (dark) and hip flexors (light). The actual number of trials are given. One regular MMT of hip flexors was rated as unclear and was excluded from this diagram.

**Figure 7 brainsci-13-01105-f007:**
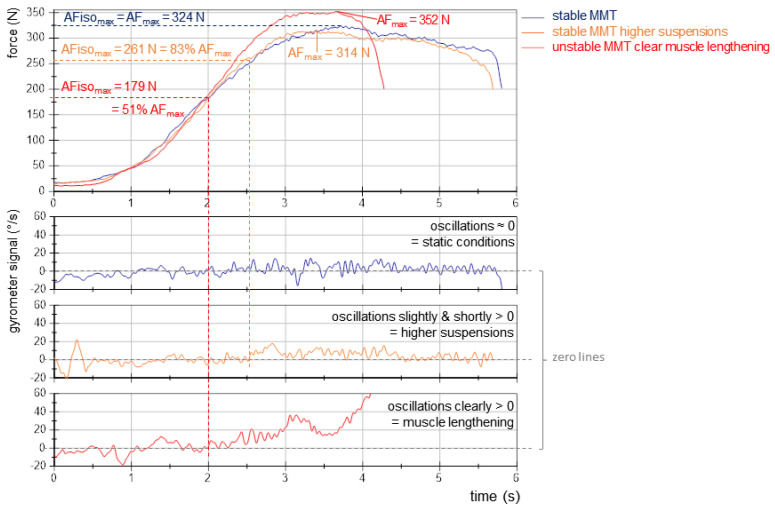
Force (above) and gyrometer signals (below) of three measurements of one participant during a clearly stable regular MMT (blue), a regular MMT which was rated as stable with higher suspensions (orange) and a clearly unstable MMT after manual spindle technique (red). The values of AFiso_max_ and AF_max_ are given according to the algorithm used. This predefines that a yielding is present as soon as the gyrometer signal increases above zero until the maximal value is reached, irrespective of the amount of increase.

**Table 1 brainsci-13-01105-t001:** Arithmetic means (M) and standard deviations (SD) of MVIC and AF parameters for regular MMT and MMT after manual spindle technique. Results of paired *t* test (*t* value, degrees of freedom (df), significance *p*, effect size Cohen’s *d_z_*) or of Wilcoxon test (standardized test statistic z, sample size n, *p* and effect size Pearson’s r) comparing regular vs. spindle are given.

Parameter	Procedure	M	SD	*t* or *z* *	df or n *	*p*	*d_z_* or *r* *
Elbow flexors							
MVIC (N)	-	313.793	40.513	-	-	-	-
AF_max_ (N)	regular	294.316	29.268	−3.213	10	0.009	0.969
	spindle	318.040	41.515				
AFiso_max_ (N)	regular	286.012	32.241	9.325	10	<0.001	2.812
	spindle	197.228	47.738				
AFosc (N)	regular	246.187	33.040	−8.950	10	<0.001	2.698
	spindle	295.412	38.131				
ratio AFiso_max_/AF_max_	regular	0.971	0.039	2.934 *	11 *	0.003 *	0.885 *
	spindle	0.614	0.101				
ratio AFiso_max_/MVIC	regular	0.919	0.103	7.340	10	<0.001	2.213
	spindle	0.626	0.123				
ratio AFosc/AF_max_	regular	0.834	0.043	−7.455	10	<0.001	2.248
	spindle	0.930	0.045				
ratio AFosc/AFiso_max_	regular	0.864	0.065	−2.943 *	11 *	0.003 *	0.885 *
	spindle	1.692	0.599				
slope (lg(N/s))	regular	2.020	0.207	−1.511 *	11 *	0.131	-
	spindle	2.069	0.205				
Hip flexors							
MVIC (N)	-	311.525	44.415	-	-	-	-
AF_max_ (N)	regular	295.816	38.929	−5.150	11	<0.001	1.487
	spindle	327.257	52.307				
AFiso_max_ (N)	regular	289.512	46.788	10.028	11	<0.001	2.895
	spindle	209.156	50.657				
AFosc (N)	regular	230.325	34.084	−9.700	11	<0.001	2.800
	spindle	275.344	25.778				
ratio AFiso_max_/AF_max_	regular	0.978	0.074	3.059 *	12 *	0.002 *	0.883 *
	spindle	0.637	0.105				
ratio AFiso_max_/MVIC	regular	0.932	0.104	9.231	11	<0.001	2.478
	spindle	0.670	0.128				
ratio AFosc/AF_max_	regular	0.779	0.044	−3.039	11	0.011	0.877
	spindle	0.851	0.063				
ratio AFosc/AFiso_max_	regular	0.805	0.084	−3.059 *	12 *	0.002 *	0.883 *
	spindle	1.398	0.292				
slope [lg(N/s)]	regular	2.097	0.083	0.038	11	0.970	-
	spindle	2.096	0.108				

* Wilcoxon test with *z* value, sample size *n* and effect size Pearson’s *r*. AF_max_ = max. Adaptive Force; AFiso_max_ = max. isometric AF, AFosc = AF at onset of oscillations.

**Table 2 brainsci-13-01105-t002:** Tester’s ratings of MMTs and corresponding AF values. Given is the number of MMTs which the tester rated as stable, unstable or unclear during the trials for elbow and hip flexors. The corresponding values (M and intraindividual CV) are displayed for the ratio AFiso_max_ to AF_max_.

Muscle	Rating Tester	Number MMTs	AFiso_max_/AF_max_	
		n	M	CV_intra_ ^1^
Elbow flexors	stable	33	0.97	0.05
	unstable	33	0.61	0.24
	unclear	-	-	-
Hip flexors	stable	35	0.99	0.02
	unstable	36	0.64	0.13
	unclear	1	0.53	-

^1^ Group arithmetic mean of the CVs of the trials of one participant (intraindividual).

## Data Availability

The data presented in this study are available in the article.

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
