# Peer review of "Another Way to Confuse Motor Control: Manual Technique Supposed to Shorten Muscle Spindles Reduces the Muscular Holding Stability in the Sense of Adaptive Force in Male Soccer Players"

_brainsci, 2023, doi:10.3390/brainsci13071105_

Round 1
Reviewer 1 Report
The authors performed an in-depth study of adaptive force parameters using a manually implemented slack procedure. Previous work of the working group has already covered this topic in detail. Although the present study contains new information, its novelty is not significant. However, the thorough evaluation, critical examination and integration of new elements into a global picture are included in this paper.
The article is easy to follow, and the description of the parameters, which at first glance may seem a little difficult to understand, is satisfactory. However, there is a noticeable amount of verbatim quotation from previous articles. In my opinion, the article also contains some minor points that need correction or explanation:
Line 160 Do the impaired neuromuscular functions mentioned above affect the outcome of the measurements? Are these injuries known or assumed?
Figure 1 in panel (b) it is barely visible what is happening between the examiner and the patient. In panels (d) and (e), it might be interesting to specify the distances between the examiner's hands.
Line 243: For the analysis of the force and gyrometer signals, only specific references are given. Are others using this handheld and similar data analysis?
Table 1: I could not find the explanation of some of the parameters in the table in the text (T or z; df or n , p), only in the table signature.
In the case of the parameters slope (lg(N/s)) for elbow and hip flexors, it is particularly curious that one shows a significant difference and the other does not (given the mean values and standard deviations).
Table 2: I find the representation of the stable, unstable and unclear categories a bit confusing. It does not distinguish which were measured following regular MMT and which following spindle technique.
Line 535: In this series of experiments, football players were tested, which could potentially affect the measurement. Apart from the untrained (which is mentioned), were any other groups of athletes tested?
Although the last section of the discussion mentions the usefulness of the observations, I would suggest that the relevance of this work to sports physiology should be emphasised a little more.
Author Response
Dear Reviewer,
Thank you for your positive reply, interest in our research and helpful comments.
We tried to address all of them – see point-by-point-response below (in blue italic) – and hope the revisions were done to your satisfaction.
One major hint we have to make: we revised the slope algorithm. The former algorithm did not capture the slope properly for very strong participants and special behavior at higher force levels especially due to the oscillations at those high force values. Hence, we adjusted the area of slope analysis to the relevant area of linear force increase (from 40 to 70% of the minimal AFisomax value). We included this in the manuscript (methods, results incl figure and table, discussion). This led also to the benefit that we did not have to exclude some trials. Consequently, it mirrors the slope much better. Furthermore, the manuscript could be shortened thereby. We hope that you can agree on the rationale behind it. Please except our apologies that we did not include this from the beginning.
Thank you for your effort!
Sincerely,
The authors
Point-by-point-response:
The authors performed an in-depth study of adaptive force parameters using a manually implemented slack procedure. Previous work of the working group has already covered this topic in detail. Although the present study contains new information, its novelty is not significant. However, the thorough evaluation, critical examination and integration of new elements into a global picture are included in this paper.
- Thank you for your feedback. Decisive is that this investigation used a manual procedure to shorten muscle spindles, which was not investigated before. Furthermore, the focus on muscle stability in the sense of AF is also novel. From our perspective this has significance - but surely, this is our opinion ;-)
The article is easy to follow, and the description of the parameters, which at first glance may seem a little difficult to understand, is satisfactory. However, there is a noticeable amount of verbatim quotation from previous articles.
- Yes, you are totally right. There is a lot of verbatim quotation. We included it because we wanted to avoid plagiarism. We revised the manuscript accordingly and tried to avoid verbatim quotation were appropriate.
In my opinion, the article also contains some minor points that need correction or explanation:
Line 160 Do the impaired neuromuscular functions mentioned above affect the outcome of the measurements? Are these injuries known or assumed?
- Thank you for this critical question. The muscles which showed an impaired neuromuscular function were excluded from the measurements. Thus, they did not affect the results. We did not take a closer look at those muscles or the reason for the impaired neuromuscular control. However, the participants stated to have complaints (after injuries or else) of the respective limbs. This was an exclusion criterion.
Figure 1 in panel (b) it is barely visible what is happening between the examiner and the patient. In panels (d) and (e), it might be interesting to specify the distances between the examiner's hands.
- You are right, it is difficult to see what happens in panel (b). We exchanged the picture. We added the distances between the examiner’s hands. Thank you for that suggestion.
Line 243: For the analysis of the force and gyrometer signals, only specific references are given. Are others using this handheld and similar data analysis?
- Since we developed the device and it is not marketable yet, other researchers are not using it. Accordingly, the data analysis is also unique.
Table 1: I could not find the explanation of some of the parameters in the table in the text (T or z; df or n , p), only in the table signature.
- Yes, we did not quote them in the main text of the manuscript. They are only relevant for the table since they are statistical values and we wanted to avoid duplicate information. I am not aware that they have to be mentioned in the main text?
In the case of the parameters slope (lg(N/s)) for elbow and hip flexors, it is particularly curious that one shows a significant difference and the other does not (given the mean values and standard deviations).
- Completely right. This is a phenomenon of the used algorithm for those very strong participants and special behavior at higher force levels. Then, the algorithm which we used until now is not appropriate and distort the slope, especially due to the oscillations at those high force values. In the meantime, we revised the algorithm and adjusted the area of slope analysis to the relevant area of linear force increase (from 40 to 70% of the minimal AFisomax value). We included this in the manuscript (methods, results incl figure and table, discussion). This led also to the benefit that we did not have to exclude some trials. Consequently, it mirrors the slope much better. Furthermore, the discussion could be shortened clearly. We hope that you can agree on the rationale behind it. Please except our apologies that we did not include this from the beginning.
Table 2: I find the representation of the stable, unstable and unclear categories a bit confusing. It does not distinguish which were measured following regular MMT and which following spindle technique.
- Correct, it does not distinguish between regular and spindle procedure. Table 2 is not about this differentiation but should reflect the MMTs which were assessed as stable or unstable. Therefore, we used this description. In the text it is mentioned if they were assessed during regular or spindle procedure. I hope you can agree on this.
Line 535: In this series of experiments, football players were tested, which could potentially affect the measurement. Apart from the untrained (which is mentioned), were any other groups of athletes tested?
- Until now the soccer players were the only homogeneous athletes’ group which we tested. The other experiments included partly sports students. However, they were not involved in professional sports.
Although the last section of the discussion mentions the usefulness of the observations, I would suggest that the relevance of this work to sports physiology should be emphasised a little more.
- From our perspective, the relevant points are included. What do you have in mind? The relevance for muscular stability with respect to risk of injury or complaints is mentioned – which from our point of view is the most relevant. We added a sentence for the relevance to consider rather muscle stability than muscle strength in the future.
Reviewer 2 Report
This manuscript aims to investigate the effect of a manual spindle technique on parameters of AF assessed by an objectified MMT compared to regular testing without manipulation in participants with a “normoreactive” muscle function. The subject is very interesting and the manuscript is well structured. I have the following suggestions:
1. The introduction is too long, needs to be revised, and should be better structured.
2. There are many strange uses of double quotes that need further modification. For example,
-Line 59-61
-Line 66-67, e.g., “if the muscle is stretched, contracted at the stretched length and held there for several seconds, stable cross-bridges will form at the longer length (…).
-Line 429-431, If “…the force produced by the tester is too low, an MMT falsely assessed as stable could be the result even though the tested muscle would be actually unstable.” [14] (p. 12).
-Line 620
...
Is this sentence copied directly from the Ref No. 20? And what does “…” mean?
3. “4.2 Limitations” is suggested to be moved to the end of the discussion.
4. Figures are recommended to be made with professional graphing software, not Excel.
5. The conclusion is too long, needs to be revised, and should be better structured.
Author Response
Dear Reviewer,
Thank you for your positive reply, interest in our research and helpful comments.
We tried to address all suggestions – see point-by-point-response below (in blue italic) – and hope the revisions were done to your satisfaction.
One major hint we have to make: we revised the slope algorithm and results. The former algorithm did not capture the slope properly for very strong participants and special behavior at higher force levels especially due to the oscillations at those high force values. Hence, we adjusted the area of slope analysis to the relevant area of linear force increase (from 40 to 70% of the minimal AFisomax value). We included this in the manuscript (methods, results incl figure and table, discussion). This led also to the benefit that we did not have to exclude some trials. Consequently, it mirrors the slope much better. Furthermore, the manuscript could be shortened thereby. We hope that you can agree on the rationale behind it. Please except our apologies that we did not include this from the beginning.
Thank you for your effort!
Sincerely,
The authors
This manuscript aims to investigate the effect of a manual spindle technique on parameters of AF assessed by an objectified MMT compared to regular testing without manipulation in participants with a “normoreactive” muscle function. The subject is very interesting and the manuscript is well structured.
- Thank you for your feedback and interest in our work!
I have the following suggestions:
- The introduction is too long, needs to be revised, and should be better structured.
- There are many strange uses of double quotes that need further modification. For example,
-Line 59-61
-Line 66-67, e.g., “if the muscle is stretched, contracted at the stretched length and held there for several seconds, stable cross-bridges will form at the longer length (…).
- We revised the introduction, removed the “strange uses of double quotes” and tried to shorten it. However, the structure seems to be appropriate to us (state of knowledge of development of injuries/complaints, relevance of muscle stability, introduction of AF, relevance of muscle spindle afferents for sensorimotor control, current research on slacked muscles, introduction of different procedure to generate a slack, assessment of AF and definition of stable and unstable muscles, aim and hypotheses of the study). We hope that you can agree to the revised version.
-Line 429-431, If “…the force produced by the tester is too low, an MMT falsely assessed as stable could be the result even though the tested muscle would be actually unstable.” [14] (p. 12).
- We revised this sentence.
-Line 620
Is this sentence copied directly from the Ref No. 20? And what does “…” mean?
- Yes, all sentences in quotes are copied directly from the given references. For some sentences it is difficult to reword them. We tried to do this were appropriate. The “…” indicate missing parts of the original quotation. I checked the correct way for English quotation – must be […]. However, we reduced the number of direct quotations and missing parts are not relevant anymore.
- “4.2 Limitations” is suggested to be moved to the end of the discussion.
- Done
- Figures are recommended to be made with professional graphing software, not Excel.
- This seems to be a question of taste and does not influence the content of the manuscript or the results. Is there a disadvantage of Excel diagrams? We would prefer to retain the figures as they are. We hope that you can agree on that.
- The conclusion is too long, needs to be revised, and should be better structured.
- We shortened and restructured it in parts. Hope this will be fine for you.